# Whole-genome sequencing in 333,100 individuals reveals rare non-coding single variant and aggregate associations with height

The role of rare non-coding variation in complex human phenotypes is still largely unknown. To elucidate the impact of rare variants in regulatory elements, we performed a whole-genome sequencing association analysis for height using 333,100 individuals from three datasets: UK Biobank (N = 200,003), TOPMed (N = 87,652) and All of Us (N = 45,445). We performed rare ( < 0.1% minor-allele-frequency) single-variant and aggregate testing of non-coding variants in regulatory regions based on proximal-regulatory, intergenic-regulatory and deep-intronic annotation. We observed 29 independent variants associated with height at $P < 6 \times 10^{-10}$ after conditioning on previously reported variants, with effect sizes ranging from −7cm to +4.7 cm. We also identified and replicated non-coding aggregate-based associations proximal to *HMGA1* containing variants associated with a 5 cm taller height and of highly-conserved variants in *MIR497HG* on chromosome 17. We have developed an approach for identifying non-coding rare variants in regulatory regions with large effects from whole-genome sequencing data associated with complex traits.

The role of rare non-coding variation in common human phenotypes is still largely unknown. Previous studies have been largely limited to studying common variations using genotyping arrays or rare variations in the coding regions of genes using exome sequencing. Studies of rare variation in the non-coding genome, which is by far the most abundant form of inherited variation, could lead to the identification of important gene regulatory elements with large effects on human diseases and traits.

Most genetic variation associated with complex phenotypes lies in non-coding regions of the genome[1]. Array-based genome-wide association studies have had substantial success at identifying common variants associated with complex phenotypes and disease[2]. For height a large proportion of the common variant heritability has been explained[2]. In contrast, the identification of rarer variation, potentially with substantially larger effects, has been largely limited to coding

variation based on exome sequencing (e.g., loss-of-function variants in *GIGYF1* associated with diabetes[3]) or imputation of lower frequency variants[2].

Despite the success of common variant and rare coding variant-based approaches, the vast majority of inherited human genetic variation is both rare and in the 99% of the genome that is non-coding. Identifying the rare non-coding variation associated with common diseases and traits could reveal new regulatory gene mechanisms, and substantially increase our understanding of human biology and disease.

Whole genome sequencing (WGS) has been successful at identifying rare non-coding causes of monogenic disease in several cases[4,5]. For example, we have recently shown that rare variants in an intronic regulatory element of *HK1* causes inappropriate expression of Hexokinase 1 in pancreatic beta-cells leading to congenital

e-mail: g.hawkes2@exeter.ac.uk; m.n.weedon@exeter.ac.uk

hyperinsulinism[6]. However, there have been few sequencing-based studies aiming to identify rare non-coding variation associated with complex phenotypes[7], despite estimates of the relative functional importance of the non-coding genome of 6–15%[8,9]. Two recent studies from TOPMed performed WGS rare-variant analysis for lipid-levels[10] (N = 66,000), where they identified suggestive associations with variants in DNA hypersensitivity sites proximal to *PCSK9* altering lipids, and in blood pressure[11] (N = 51,456), where genomic aggregate signals at *KIF3B* were identified.

Identifying associations between rare variants and complex traits has several advantages over common variant associations. Firstly, rare causal variants are likely to have larger effect sizes and so potentially be of greater clinical relevance. Secondly, rare variants are less likely to be in linkage disequilibrium with other variants and so provide more direct information about likely causal regulatory regions and genes involved. Finally, rare variant aggregate associations, where genetic variants of similar predicted consequence and location are tested in aggregate, can also provide strong evidence for specific non-coding elements that are responsible for an association compared to single variant associations.

We performed an analysis of height, a model complex trait, focussed on identifying novel rare variant associations from large-scale WGS data. We performed a discovery analysis using WGS data on 200,003 individuals from UK Biobank (UKB) and replicated our results in 133,097 individuals from All of Us[12] and TOPMed[13]. We show that our approach can identify rare single variant and aggregate associations in the non-coding genome that have not been previously identified. Importantly, our analytical approach to WGS-based association analyses can be applied to other complex phenotypes.

We performed discovery association analyses using WGS data on 200,003 individuals from the UKB, a population cohort from the United Kingdom[14]. We analysed rank inverse-normalised standing height, a model polygenic trait, with genomic data on 789,700,118 genetic variants including single nucleotide variants (SNVs), small insertions/deletions (indels) and large structural variants (SVs) including copy number deletions and duplications. To identify rare non-coding genetic associations that have not been previously identified, we conditioned our analyses on 12,661 variants from the latest GIANT height consortium analysis of 5.4 million people[2] based on imputed genotype array data, an exome-array analysis of height[15], and genome-wide significant ($P < 5 \times 10^{-8}$) variants from an exome-wide association study of height[16]. Our primary discovery analysis was performed in 183,078 individuals of genetically-inferred European ancestry. We also performed the same analyses in individuals with genetically-inferred South Asian (N = 4439) and African (N = 3077) ancestry in the UK Biobank. We replicated our results in a cross-ancestry analysis using 87,652 individuals with WGS from TOPMed, and 45,445, 20,548 and 13,683 individuals with genetically-inferred European, African and self-reported Hispanic ancestry/ethnicity with WGS data in All of Us respectively (refer to Supplementary Data 1 for a breakdown of ancestries and cohort demographics). Statistical significance for single variants was defined as $P < 6.3 \times 10^{-10}$, and $P < 6.58 \times 10^{-10}$ for genomic aggregates, based on 20 simulated randomly generated phenotypes (see "**Methods**"). Association statistics are based on a two-sided chi-squared test, unless otherwise stated.

## Single Variant Association Testing
We tested all genetic variants with a minor allele count (MAC) ≥ 20, excluding variants with a low-quality genotype calling score (graphTyper AA score <0.5), using REGENIE[17]. Variants which were associated at the stated statistical threshold were then clumped using PLINK[18], and a sequential variant conditioning procedure was applied to determine the variant most likely to be responsible for the signal (see "**Methods**").

## Genomic Aggregate Association Testing
After annotating each variant using the Ensembl Variant Effect Predictor[19], we segmented variants in the genome into classification groups, including gene-centric (i.e., coding and splicing; or proximal regulatory, including 5 kb upstream and 5 kb downstream - +/− 5 kbp from the 5/3′ UTR's) and non-gene-centric potentially regulatory variation (intergenic and intronic based on any transcript), as well as a sliding window test that covered the whole genome, excluding exons. We performed genomic unit aggregate testing limited to rare (within-sample minor allele frequency, MAF < 0.1%) genetic variants in functionally annotated regions based on three published weights representing in silico predicted deleteriousness (Combined Annotation Dependent Depletion, CADD[20]), conservation (Genomic Evolutionary Rate Profiling, GERP[21]) and non-coding constraint (Junk Annotation Residual Variation Intolerance Score, JARVIS[22]). Variants that were classified as coding in any transcript were excluded from regions we defined as proximal (within 5kbp of the 5/3′ UTR[19]), and variants in proximal regions were subsequently excluded from regions defined as non-proximal potentially regulatory regions−see the "**Methods**" section for precise definitions. We refer to proximal-regulatory regions and non-proximal regulatory regions as "proximal" and "regulatory" respectively for the remainder of the manuscript.

## Results

### We identfied 29 rare and low-frequency not previously identified single variants associated with human height in UKB
After adjusting for published height genetic variants (Supplementary Data 2), 28 rare (MAF < 0.1% & MAC > 20) and low-frequency (0.1% < MAF < 1%) SNVs and indels remained independently associated with height (Fig. 1). These variants had effect sizes ranging from −7.25 cm to + 4.71 cm (−0.79 to 0.52 SD)−see Supplementary Data 3 & 4. As expected, variants with a lower minor-allele-frequency had the largest effect estimates (Fig. 2).

We additionally identified evidence of association with a 47,543 bp structural deletion in the pseudo-autosomal region of chromosome X (X:819,814-867,357). The proximal-*SHOX* deletion occurs 173kbp downstream of *SHOX*, and is present in 0.3% of the population and associates with lower height (β = −2.79 cm [−3.33, −2.25], $P = 5.01 \times 10^{-24}$, Supplementary Data 3). This exact deletion, downstream of *SHOX*, has only previously been reported in clinical cohorts with Leri-Weill dyschondrosteosis[23], a genetic disorder characterised by shortened limbs and short stature. In these clinical cohorts, 15% had at least one copy of the 47.5kbp deletion. In the UKB population, the deletion was present in 824 individuals (0.3%) (one carrier was a homozygote).

### Three rare single variant associations showed robust evidence of replication in TOPMed and All of Us
Twenty two of the 28 low-frequency SNVs/indels we identified had consistent sign (binomial $P = 1.51 \times 10^{-3}$), and ten showed nominal ($p < 0.05$) evidence of replication in a meta-analysis of TOPMed and All of Us when we would expect 1-2 (1.4 expected at $P = 0.05$)−Supplementary Data 5. Three loci replicated at Bonferroni significance ($P < 0.05/27$) in a meta-analysis of the replication datasets. We estimate that in the replication dataset we are powered (power > 80%) to replicate 9 signals at $P < 0.05$ and 4 at $P < 1.85 \times 10^{-3}$. These variants were in the promoters of *HMGA1* (6:34237902:G:A, β = 4.71 cm [3.41, 6.01 cm], $P = 1.29 \times 10^{-12}$, replication $P = 6.82 \times 10^{-7}$), *GHRH* (20:37261871:G:A, β = 1.82 cm [1.43, 2.23 cm], $P = 2.52 \times 10^{-19}$, replication $P = 3.13 \times 10^{-5}$), and proximal to *CUL3* (2:224492608:T:C, β = 2.72 cm [2.24, 3.19 cm], $P = 4.29 \times 10^{-11}$, replication $P = 4.20 \times 10^{-4}$). Chromosome X data was unavailable for replication.

We did not identify any novel replicating associations in the South Asian or African ancestry-specific analyses in the UKB. Only one

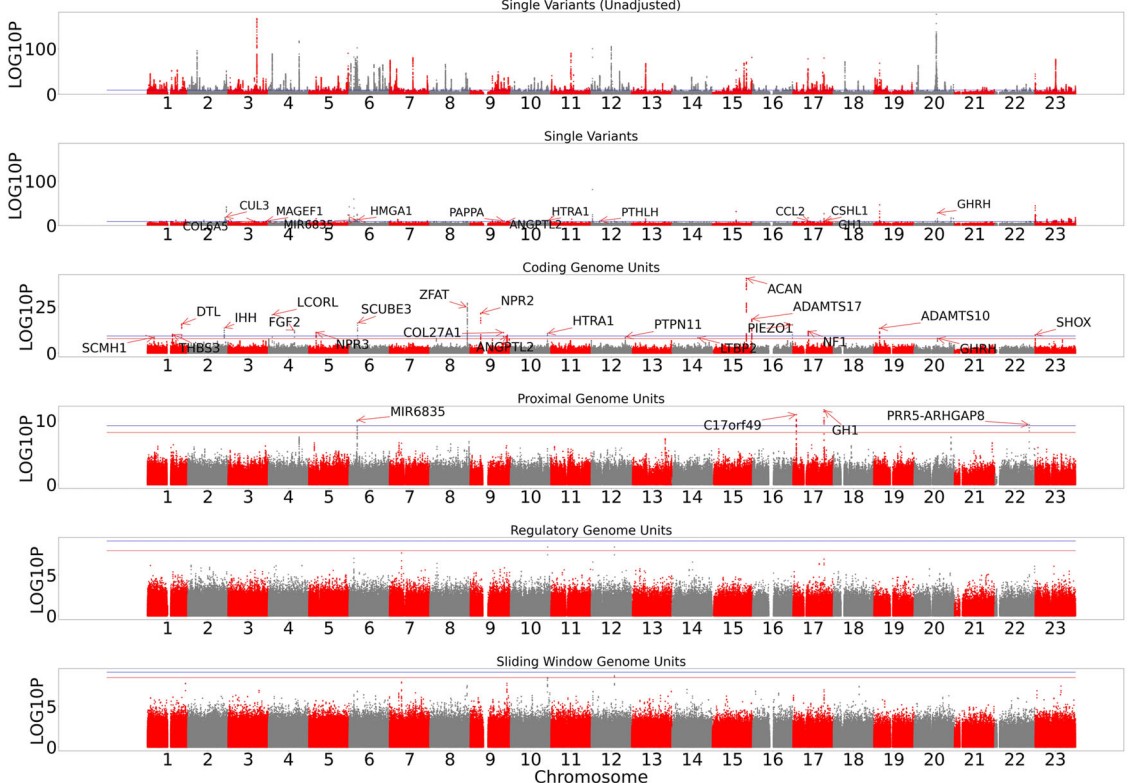

**Fig. 1 | Manhattan plots of a whole-genome sequencing analysis of height.** Manhattan plots of results split by single variant and genomic aggregate analysis. From top to bottom: unconditioned single variants, single variants conditioned on known height loci, rare ( < 0.1% minor-allele frequency) coding genome aggregates, followed by rare non-coding genome units proximal genome aggregates, regulatory genome aggregates and sliding window aggregates. We plot −log10(p) on the y-axis. Red horizontal lines indicate the position of genome-wide significance considering only that panel, whilst blue indicates genome-wide significance across the entire study. For the single variant, coding and proximal panels, loci leads are labelled by their annotated gene based on the output of the Variant Effect Predictor. All plotted statistics were derived from the discovery UK Biobank analysis set (N = 183,078), based on a two-sided chi-squared statistic.

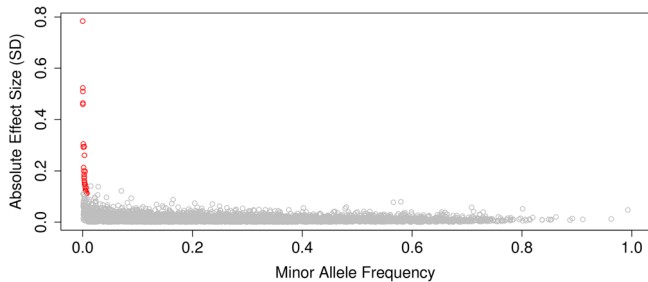

**Fig. 2 | Comparisons of rare variant effect sizes with known common effects.** Variant minor-allele-frequency versus absolute effect size for the 28 genetic variants (red) identified after adjusting for previously published height loci (derived from the discovery UK Biobank analysis set; N = 183,078), contrasted against the results of Yengo et al. [2] for common variants (grey).

genetic variant achieved genome-wide significance in an analysis of individuals of South Asian ancestry (X:116495780:AGTGTGTGTGTGT:A, $P = 1.43 \times 10^{-10}$), but did not associate in the European ($P = 0.91$) or African ($P = 0.98$) ancestry-specific analyses (we were unable to test this variant in All of Us or TOPMed).

### We identified and replicated three rare (MAF < 0.1%) non-coding regions associated with height

We performed 57,608,498 genomic aggregate association tests, consisting of 5,941,548 coding, 13,005,638 proximal regulatory, 4,861,759 intergenic/deep intronic and 33,799,553 non-coding sliding window association tests. We performed three different types of statistical test: i) 'BURDEN', where the direction of effects for all variants is assumed to be the same, ii) 'SKAT', where there is no assumption about directionality or similarity of magnitude of effects, and iii) 'ACAT', where there is no assumption about directionality or magnitude of effects and not all variants need be associated with the outcome[24].

We identified seven (partially overlapping) non-coding regions of interest based on aggregate tests ($P < 6.31 \times 10^{-10}$; Supplementary Data 6 & 7). Four regions remained significant after adjusting for previously identified height loci (Table 1). The four regions consisted of nine genomic aggregate tests proximal to: *HMGA1, C17orf49, GH1, CSHL1, PRR5-ARGHGAP8* and *MIR6835*. We did not find any novel genomic unit associations based on African or South Asian ancestry-specific analyses in our discovery analysis. We also did not observe evidence of study-wide significance for any intergenic-regulatory or sliding window aggregate associations.

The aggregate-based tests at *HMGA1* and *C17orf49* replicated in All of Us and TOPMed when combined with genetically inferred individuals of South Asian (SAS) and African (AFR) in the UKB, and we were unable to test *PRR5-ARHGAP8* for replication in non-UKB analyses, as they did not annotate fusion transcripts, and the aggregate showed no evidence of replication in the UKB-AFR (SKAT $P = 0.195$) or UKB-SAS (SKAT $P = 0.452$) analyses (Table 1).

We then performed a final analysis additionally adjusting aggregate-based tests for variants identified in our single variant analysis. Two non-coding aggregate associations remained genome-wide significant: *C17orf49* (downstream, *GERP > 2*, $\beta = 1.34$ cm [95% CI 0.931, 1.66], $P = 2.00 \times 10^{-11}$) and *PRR5-ARHGAP8* (upstream, *JARVIS > 0.99*, $P = 4.27 \times 10^{-10}$).

**Table 1 | Significant rare (<0.1%) non-coding genomic aggregate associations with human height after adjusting for known height loci ('log10p conditioned')**

| CHR | START (b38) | END (b38) | CLASS | GENE/UNIT | Annotation | TEST | BETA (SD) | SE (SD) | p | p conditioned | p conditioned + | Replication P-Value |
|---|---|---|---|---|---|---|---|---|---|---|---|---|
| 6 | 3423791 | 34238791 | Proximal | HMGA1 | Upstream & Upstream (JARVIS > 0.99) | ACAT | NA | NA | 1.58E-11 | 1.55E-10 | 3.72E-07 | 0.00183[a] |
| 17 | 7017304 | 7023304 | Proximal | C17orf49 | Downstream (GERP > 2) | BURDEN | 0.14 | 0.02 | 1.26E-11 | 1.26E-11 | 2.00E-11 | 1.40E-02 |
| 17 | 63918839 | 63923839 | Proximal | GH1 | Upstream (GERP > 2) | BURDEN | −0.33 | 0.05 | 5.01E-12 | 2.00E-12 | 4.27E-04 | 2.86E-01 |
| 22 | 44809805 | 44814805 | Proximal | PRR5-ARHGAP8 | Upstream (JARVIS > 0.99) | SKAT | NA | NA | 3.72E-10 | 4.37E-10 | 4.72E-10 | NA |

The CLASS column denotes how variants were classified according to Fig. 1, and the annotation column denotes how the variants were additionally grouped together (see "methods") based on variant scores. Replication was calculated as a meta-analysis of TOPMed, All of Us and non-EUR analyses within UKB. [a] Indicates that the meta-analysis was calculated using the ACAT p-value combination method, as betas are not produced for the ACAT aggregate tests.

## Multiple rare variants, and a common variant, form an allelic series in a regulatory region upstream of HMGA1, with substantial effects on height

There were 2,006 rare variants included in the upstream non-coding association for *HMGA1* (High-mobility group protein) in the UKB, 603 of which had MAC ≥ 5 – Supplementary Data 1. Several variants appeared to be responsible for these aggregate signals (Fig. 3). The two rare variants most strongly associated with increased height were 6:34237902:G:A ($\beta$=4.83 cm, $P = 2.00 \times 10^{-13}$, MAF = 0.04%) and 6:34236873:C:G ($\beta$ = 3.97 cm, $P = 1.00 \times 10^{-10}$, MAF = 0.0470%). The five most-strongly associated variants, at $P < 5.76 \times 10^{-6}$ (Fig. 3C), were statistically independent of each other, as determined by sequential conditional testing. Our results remained statistically significant after removing several low-quality indels ($P = 1.45e-11$).

The most strongly associated rare variant alters the first base of the transcription start site of the MANE Select transcript (ENST00000311487.9, NM_145899.3) of *HMGA1*[25] (Fig. 3). This variant could result in reduced transcription of this transcript and may result in an alternative start site becoming dominant.

The next four most-strongly associated variants clustered in two adjacent enhancers in the promoter region of *HMGA1* (Fig. 3A). We also fine-mapped a previously reported GWAS signal to the same enhancer (6:34237688:G:GGAGCCC, MAF = 10.9%, $P = 6.50 \times 10^{-103}$), with posterior probability > 0.99 and 95% credible set of size 1 (Fig. 3B).

We next searched for evidence of a role for coding variation in the impact of *HMGA1* in height. *HMGA1* is a constrained gene (pLI score = 0.83) and there are no predicted protein truncating variants in the UKB and only a single individual with a first exon frameshift in gnomAD. There was also no evidence of individual missense or aggregate coding association with height for *HMGA1* either in the UKB WGS data (min($P$) = 3.09e-4), or in *GeneBass*, based on 394,841 exome-sequences from UKB (min($P$) = 0.284).

## Rare variants of microRNA host-gene MIR497HG affect height

There were 235 highly conserved (GERP > 2) rare variants which contributed to the non-coding *C17orf49* (Chromosome 17 Open Reading Frame 49) genomic aggregate result in the UKB, cumulatively associated with a 1.36 cm increase in height (95% CI 1.11, 1.48 cm, $P = 1.26 \times 10^{-11}$), 59 of which had MAC ≥ 5 – see Supplementary Data 9. Of the 235 variants that contributed to the aggregate signal, 152 (64.7%) had an effect estimate with the same direction of effect as the aggregate (binomial $P = 7.96 \times 10^{-6}$), suggesting that multiple variants are responsible.

The proximal region of *C17orf49* overlaps with microRNA host cluster *MIR497HG*, from which microRNAs *MIR195* and *MIR497* are derived (Fig. 4A). We thus re-analysed the *C17orf49* proximal region excluding miRNA variants, and additionally tested the microRNA as independent genome units (Fig. 4B). The strength of association between the identified *C17orf49* proximal aggregate and height was reduced after removing any variant overlapping miRNA ($\beta$ = 1.11 cm, $P = 3.98 \times 10^{-5}$)–Supplementary Data 10. Further, the association between a genomic aggregate of miRNA variants in *MIR195* and height was more than double that of the primary *C17orf49* signal ($\beta$ = 3.05 cm [95% CI 1.44, 4.65 cm, $P = 1.97 \times 10^{-4}$]), showing nominal evidence of heterogeneity ($P = 0.0454$) to the primary signal.

The variants contributing to the *MIR497HG* signal occurred in the promoter region and in the two 2 miRNA products, *MIR195* and *MIR497*. This suggests the possibility of two mechanisms that contribute to the association−variants altering the expression of the host gene *MIR497HG*, and variants specifically affecting the miRNA sequence.

There is extensive literature on the genes that *MIR195* and *MIR497* bind and affect expression of, while there is little previous literature referencing *C17orf49*, except for a small number of studies of cancer phenotypes[26]. *MIR497* and *MIR195* expression have been associated with a range of genes that influence cancer[27], and both have been

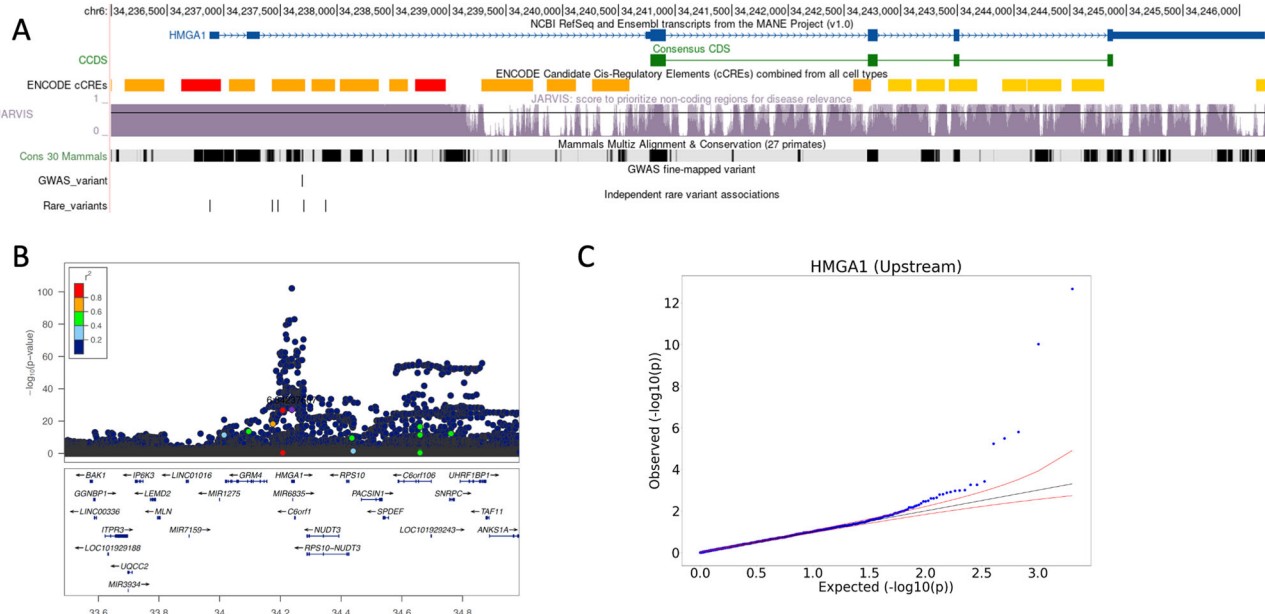

**Fig. 3 | Identification of a regulatory region associated with height proximal to HMGA1. A** UCSC genome browser window showing genomic features in the region upstream of HMGA1, including JARVIS score, conservation score, known ENCODE cCRE's and consensus coding sequence. Custom track 'Common Variants' shows the locations and −log10(P) values of variants with MAF > 0.01%, and 'Rare Variant Associations' displays the locations and −log10(P) values of variants which contributed to the genomic aggregate (MAF < 0.001%). **B** Manhattan plot showing the distribution of log10-pvalues centred on the common GWAS signal at the HMGA1 locus. **C** QQ-plot of −log10(P) values for variants which were included in the aggregate test. All plotted statistics were calculated from the discovery UK Biobank analysis set (N = 183,078) based on a two-sided chi-squared statistic.

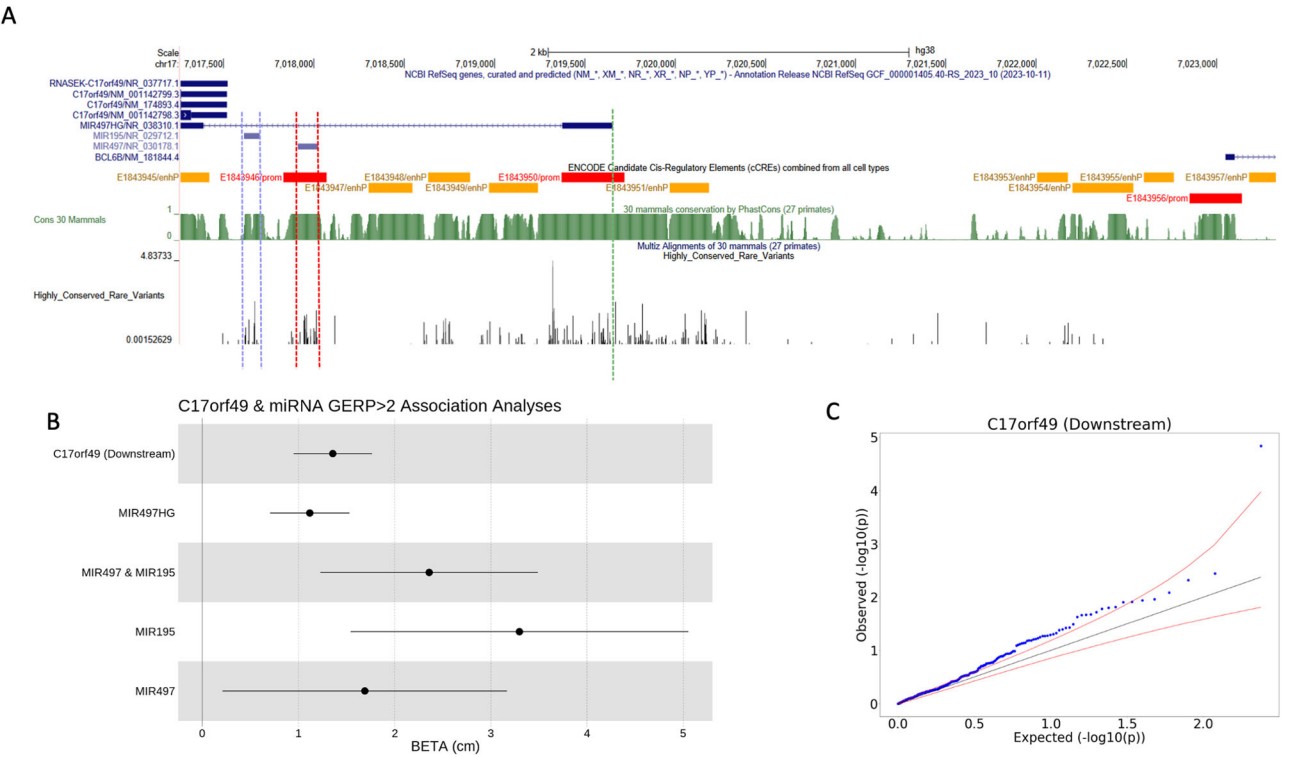

**Fig. 4 | Identification of a regulatory region associated with height proximal to C17orf49, overlapping a miRNA. A** UCSC genome browser window showing genomic features in the region of the region upstream of C17orf49, including JARVIS score and conservation score. −log10(P) values of rare ( < 0.01%) variants which contributed to the aggregate association are highlighted in a custom track based on a two-sided chi-squared test statistic. The vertical blue, red and green lines show the boundaries of MIR195, MIR497 and MIR497-HG respectively. **B** Forest plot demonstrating how the effect estimate for the association between the proximal and miRNA aggregates, depending on how variants are allocated. Error bars show the standard error of the effect size estimate. **C** QQ plot for variants in the C17orf49 proximal aggregate. All plotted statistics were calculated from the discovery UK Biobank analysis set (N = 183,078) based on a two-sided chi-squared statistic.

implicated in quiescence of skeletal muscle cells[28]. Reduced expression of *MIR497* has also been shown to promote osteoblast proliferation and collagen synthesis[29]. Zhao et al. also reported an association between down-regulation of *MIR497*, one of the three miRNA which overlapped with the proximal aggregate, and idiopathic short stature in a clinical cohort of Chinese children with short stature[30] ($P < 0.05$). Zhang et al.[31] have additionally implicated *MIR497* in chondrogenesis (cartilage development), and shown that the miRNA impacts *IHH* (Indian Hedgehog Homologue), which is essential for bone formation[32].

*MIR195* has also been shown to interact with *HMGA1* and affect expression. For example, it has been shown that *MIR195* and *MIR497* repress *HMGA1*, which in turn downregulates one of the HMGA1 downstream targets Id3, which has an inhibitory effect on myogenic differentiation[33]. We therefore tested interaction of the common *HMGA1* variant and the miRNA, but did not detect an association either at the aggregate (min($P$) = 0.541) or single variant (min($P$) = 3.09 × $10^{-3}$) level.

### Promoter variants of GH1 have substantial effects on height
Nine rare highly conserved (GERP > 2) variants contributed to the upstream non-coding aggregate for *GH1* (Growth Hormone 1) in the UKB, 5 of which had MAC ≥ 5—see Supplementary Data 11. The aggregate signal was associated with a 0.34 SD (3.11 cm) reduction in height. One of the 9 variants, which replicated, was independently associated with height (17:63918961:A:G, β = −4.24 cm [95% CI −5.53, −2.94 cm], $P = 1.46 × 10^{-10}$, MAF = 0.04%), and has previously been reported as a variant of unknown significance in multiple clinical cohorts for idiopathic short stature as NM_000515.5_c.185 T > C. These findings in clinical cohorts of idiopathic short stature include: three carriers of the variant we identified (c.185 T > C) were previously identified in a Sri Lankan cohort of patients with Isolated Growth Hormone deficiency[34] (IGHD); three siblings with consanguineous parents with IGHD[35]. The variant was originally identified in a cohort of 41 unrelated children with short stature, and 11 unrelated patients with IGHD[36] (as −123T > C). That study showed that that the variant occurs in a distal binding site for *POU1F1*[36] (Pituitary-specific positive transcription factor 1), which might regulate *GH1*[37].

## Discussion
By conducting one of the largest whole-genome sequence-based analyses to date with a focus on rare non-coding variation, we have provided novel insights into the genetic architecture of height not previously detected by standard array-based GWAS or exome sequencing approaches. Our results clearly demonstrate that our approach to analysing whole-genome sequencing data has revealed a largely untapped potential for linking rare non-coding genetic variation to complex, common human phenotypes.

We identified six non-coding regions based on genomic aggregate testing, four of which contained at least one genomic aggregate that survived adjustment for genetic variation known to impact height. We presented evidence for replication of three of these non-coding genomic aggregates, proximal to *C17orf49*, *GH1* and *HMGA1*. These loci implicated highly-conserved miRNA regulating gene expression, an altered transcription start site, pituitary growth factor co-gene regulation, multiple proximal enhancers, and conservation and constraint of genetic variation in the biology of human growth via height.

We additionally found evidence of 23 low-frequency (0.1% < MAF < 1%) and 5 rare (MAC > 20 and MAF < 0.1%) single variants, after conditioning on all previously published variants. Three of the variants identified were proximal, non-coding associations (*CUL3*, *HMGA1*, *GHRH*) that showed strong evidence of replication in the All of Us and TOPMed studies.

Our work further highlights the importance of adjusting for common variants in rare and low frequency variant discovery analyses to circumvent linkage-driven associations. Before adjustment for common variants, we observed 319 rare and low frequency variants, which dropped to 80 (non-independent) after adjusting.

We chose to report genetic aggregate results after correcting for known variation only, despite some genes (e.g., *HMGA1*) containing genetic variants that were independently significant in our analysis. Although conditioning upon independent variants within the aggregates often decreased the strength of association, we do not interpret this as a suggestion that the association at the locus is driven entirely by a single variant. This is a topical point for rare variant analyses: at sufficiently high sample sizes, we predict that a large proportion of genetic variants within an identified genetic aggregate will be independently associated. We propose that this does not imply, however, that the association itself is not aggregate.

There are some limitations to our study. First, we acknowledge that our study is currently limited by sample size: a maximum allele frequency cut-off of 0.1% for genomic aggregate restricts our analysis to approximately 183 carriers per variant. Upcoming releases of whole-genome sequencing data from UKB, All of Us and TOPMed will substantially increase the identification of novel findings. Sample sizes for analysis of individuals not of inferred European genetic ancestry were particularly limited, restricting rare variant analysis and reducing statistical power more so than for common variant analysis. We were additionally limited to replication in non-UKB datasets: future methodological advances will allow individual-level meta-analysis, substantially increasing statistical power. However, this should not understate the significance of the replication of our findings in independent cohorts with differing ancestral backgrounds. Further, there is a lack of high-quality tissue-based functional data available for the non-coding genome, which will improve as more non-coding sequencing data becomes available.

In conclusion, we have identified several non-coding single variants and genomic aggregate genetic loci associated with human height using generalised annotation criteria. Our approach provides a template for future rare-variant analyses of whole-genome sequencing data of other complex phenotypes.

## Methods
### UK biobank and whole genome sequencing
The whole genome sequencing performed for UKB had an average coverage of 32.5X, with a minimum of 23.5X, using Illumina NovaSeq sequencing machines provided by deCODE[38]. The genome build used for sequencing was GRCh38: single variant nucleotide polymorphisms and short 'indels' were jointly called using GraphTyper[39]. deCODE found that the number of variants identified per individual was 40 times larger than that found using WES in the initial 150,000 releases of whole genome sequences. Structural variants were called using the same process.

Of the 200,000 individuals whose genomes were sequenced, we found, using genetic principal components as previously described[40], there were 183,078 individuals of European ancestry in this subset of the UK Biobank.

### Genetic data format
We performed a multi-allele splitting procedure on each of the 60,648 pVCF whole genome sequencing files provided by the UK Biobank using bcftools[41] and then converted those pVCFs to PLINK[18] (v1.9).bed/bim/fam format. We then grouped multiple PLINK files together, to produce 1196 non-overlapping PLINK files each covering approximately 2.5Mbp of the genome, which we use as input to REGENIE[17] (v3.1) to perform both single variant and genome unit testing.

### Common variant conditioning
We adjusted for all known loci at most 5Mbp from each variant by further grouping each of the 1196 *PLINK* format files into triplets, with

the two genotype files up- and downstream of the central *PLINK* file, to ensure that a genetic variant which was close to the beginning of an individual genome chunk was conditioned on sufficiently distant loci. We merged genome chunks at the beginning and end of a chromosome, and at either end of the centromere with only one chunk, be it downstream or upstream as appropriate.

### Genetic variant exclusion

We excluded all variants from our association analyses if *GraphTyper*, the software used to by the UK Biobank to perform genotype calling, assigned an *AAScore* which was less than 0.5[38], denoting variant quality.

### Single variant association testing

We performed single variant association testing on any variant with at least 20 carriers in the population (MAC ≥ 20, equivalent to a MAF ≥ $5.46 \times 10^{-5}$) due to the instability of regression estimates for variants with very low minor allele count[42]. We conditioned our association tests on all common variants identified in the most recently published GWAS[2] as well as published exome array variants[15], and significant ($P < 5.00 \times 10^{-8}$) exome variants published by Regeneron for standing height[16], to minimise the likelihood that our non-coding associations were driven by known common GWAS or coding loci – Supplementary Data 12.

### Null association model

We randomly generated and performed association testing for 20 normally distributed (mean zero and unit standard deviation) 'dummy' phenotypes, with an N matching that of our European ancestry analysis, in order to estimate the number of independent tests, because Bonferroni correction is known to be over-conservative for highly correlated tests. To determine a significance threshold, we took the minimum p-value across all single variant and genomic unit tests across any of the 20 simulated phenotypes, representing a 95% significance level relative to the null.

### Defining independent variants

Single variants which passed genome-wide significance were analysed using PLINK's clumping procedure, based on $r^2 < 0.001$ (linkage disequilibrium) and a minimum clump distance of 250 kb. Variants classified as independent by PLINK then underwent a formal conditional analysis step. For each window (as defined above) containing more than one 'clumped' variant, we conditioned on the top variant in the window, which we classify as an independent variant.

### LocusZoom

We generated a LocusZoom[43] plot for each genetic variant which passed our clumping procedure, based on statistical linkage disequilibrium derived from the UK Biobank whole genome sequencing data. In these cases, all variants with MAC ≥ 1 within +/− 750 kbp of the lead variant were tested for association with height, and the lead variant within the locus was determined using the PLINK clumping procedure with a maximum $r^2 \leq 0.001$ and distance of at least 250 kbp. If a variant passed only one of these criteria, we performed a bespoke independence test, where significant variants are conditioned on one-by-one until no association remains.

### Genetic variant annotation

We annotated all genetic variants using Variant Effect Predictor (VEP)[19]. Where possible, we assigned each variant to one of three *classifications*: coding, proximal-regulatory or intergenic-regulatory. A variant was classified as coding if it had an impact on an exon of **any** transcript; proximal-regulatory if the variant lay within a 5kbp window around a transcript, and was not already a coding variant in any transcript, and finally intergenic-regulatory if the variant fell within a conserved, constrained, intronic or non-coding exon region (details below), and

was neither proximal-regulatory or coding. We additionally tested variants in sliding windows of size 2000 base pairs, regardless of the number of variants in each window, with proximal and coding variants excluded to minimise hypothesis overlap.

We then assigned each variant to groupings, which we refer to as *masks*, according to their predicted consequence and location. We used five published variant scores to group variants by consequence:

### Genomic evolutionary rate profiling (GERP).

The GERP score is a measure of conservation at the variant level[21]. We classified a variant if it had a GERP score > 2.

### phastCons score.

phastCon is a window-based measure of conservation across species[44]: either strictly mammalian (phastCon 30), or for all species (phast_100). We tested non-coding genome windows, i.e., excluding any window containing an exon, that had a phastCon score in the top percentile.

### Constrained score.

Constraint was calculated in windows of size 1kbp[8] based on the local mutability and observed mutation rate of each window. We tested windows with a constraint z-score greater than or equal to four.

### Splice AI (AI) score.

The splice AI score[45] is a measure of how well predicted each variant within a pre-mRNA region is of being a splice donor/acceptor, or neither. A variant was classified as a splice site with high confidence if it had an AI > 70.

### Combined Annotation Dependent Deletion score (CADD).

The CADD score[20] predicts how deleterious a variant is likely to be. We applied the CADD score only to coding variants and considered loss-of-function variants only if tagged as high confidence by VEP. Missense variants with CADD > 25 were segregated for testing in a separate mask.

### JARVIS Score.

The JARVIS score was derived to better prioritise non-coding genetic variation for association study, based on a machine learning model derived from measures of constraint[22].

Each genome mask consisted of a number of variants with different *consequences*, based on their location, one of the above scores and/or predicted coding consequences. For example, for a variant to be classified as missense CADD > 25, it must change a codon of an exon of a gene transcript, and be predicted to be highly deleterious.

In Supplementary Data 13 we present the full list of consequences assigned to each mask and classification.

We re-assigned variants that fulfilled two distinct criteria within a given genome unit to avoid duplication. In these cases, a variant was re-labelled as a combination of the two criteria, and were attached to any mask which selects variants from at least one of those criteria.

### Pseudogenes

We assigned variants to pseudogene transcripts if they contained pseudo-exons. However, pseudo-exons were not excluded from proximal regions of non-pseudo gene associations, instead being tested as a regulatory genome unit. If a pseudo-exon overlapped with any significant genome unit signal, we performed a bespoke analysis.

### Association testing

All association analyses were corrected for age, sex, age squared, UK Biobank recruitment centre (as a proxy for geography) and the first forty genetic principal components. To account for relatedness and genomic structure, we first ran Step 1 of REGENIE[46], which generates a background null-association model for each participant and each chromosome, using 487,558 genetic variants extracted from the UKB array genotypes, after LD-pruning and frequency filtering.

## Genome unit testing

Genome unit testing was performed for variants with a maximum allele frequency threshold of 0.1%, using REGENIE, based on the genetic units specified in Supplementary Data 13. REGENIE performs four types of genome unit tests:

1. Standard BURDEN tests, under the assumption that each variant in a given gene unit mask has approximately the same effect size and sign on the phenotype
2. SKAT tests, where the sign of association of each variant in the unit is allowed to vary
3. ACAT tests, where the sign of association of each variant in the unit can differ, and only a small number of variants in the mask need be associated at all
4. ACAT-O, which is an omnibus test of BURDEN, SKAT and ACAT to maximise the statistical power across the three tests

We performed each of the four statistical tests above for each mask for which a genome unit has at least one variant. Additionally, a singleton association test was performed for all variants with MAC = 1 in each unit. REGENIE also estimated an 'all-mask' association strength for each genome unit, which is an aggregation of the test statistics of the individual masks. To ensure that this did not result in a mixing of non-coding and coding association statistics, we split each gene transcript into a coding transcript, which we tested for all coding masks, and a proximal transcript that we tested for all proximal masks. Regulatory genome units were either classified by their ENSR assignment, by the extent of a 1 kb constrained window, or a phastCon conserved window. We named sliding windows by the range of chromosomes which they covered.

## Signal Classification

We determined whether a genomic unit signal was the result of the net effect of many variants of similar consequence or driven by one variant/a single loci of variants, by performing a second batch of genomic unit association testing corrected for single variants that passed the significance threshold in the single variant analysis.

## Fine mapping

To calculate the credible set for any common variant which lay within our rare-variant loci (single variant or aggregate), we performed a fine-mapping procedure using the recently-released SuSiEx[47] software. SuSiEx leverages linkage-disequilibrium information across ancestries. $R^2$ between all variants was calculated directly from UKB WGS data, stratified by genetically determined ancestry.

## Heterogeneity calculations

We used the R-package *metafor*[48] to calculate all heterogeneity p-values between effect estimates, under the assumption of a fixed-effects model.

## Replication within non-European UKBB ancestries

We first attempted to replicate our results by repeating our analysis for individuals of South Asian (SAS) and African (AFR) ancestry, with sample sizes of 4439 and 3077, respectively.

## Replication using TOPMed

We have conducted a mutual-replication analysis with TOPMed ("Trans-Omics for Precision Medicine"), who have analysed TOPMed WGS data using the STAARpipeline[42,49,50] programme. The National Institutes of Health and the National Heart Lung and Blood in the US sponsored the creation TOPMed. The WGS was performed at a target depth of >30x using DNA extracted from blood. We analysed 87,652 multi-population samples from 33 studies in the freeze 8 TOPMed (Supplementary Data 1). The population group was defined by self-reported information from participant questionnaires in each study

(Supplementary Note). For individuals who had unreported or non-specific population memberships (e.g., "Multiple" or "Other"), we applied the Harmonised Ancestry and Race/Ethnicity (HARE) method (Fang et al. 2019; Zhang et al. 2023) to infer their group memberships using genetic data. The population groups were thus labelled by their self-identified or primary inferred population group. Among the 87,652 participants, 52,519 (60%) were female and 44,846 (51%) were non-European. Additional descriptive tables of the participants are presented in Supplementary Data 1.

## Replication using All of Us

We have also conducted a mutual-replication analysis with short-read WGS data from All of Us freeze 6, stratified by continental genetic ancestries European (EUR), AFR, and Admixed American (AMR). The All of Us team pre-computed principal components by projecting All of Us into the same PC space as the Human Genome Diversity Project and 1000 Genomes. These PCs were then used as input into a random forest classifier to derive continental ancestry classifications. Low-quality variants were removed from the dataset before association analyses were performed using REGENIE[17].

## Reporting summary

Further information on research design is available in the Nature Portfolio Reporting Summary linked to this article.

## Data availability

Data cannot be shared publicly because of data availability and data return policies of the UK Biobank. Data are available from the UK Biobank for researchers who meet the criteria for access to datasets to the UK Biobank (http://www.ukbiobank.ac.uk). Summary statistics are available at the GWAS Catalogue under accession numbers GCST90446475 (single variants) and GCST90446476 (aggregates).

## Code availability

Analysis code relating to the analyses is available via github (https://github.com/ExeterGenetics/WGS_200k_HEIGHT/tree/main).

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

## Acknowledgements

This manuscript is part of the Stratification of Obesity Phenotypes to Optimise Future Obesity Therapy (SOPHIA) project. SOPHIA has received funding from the Innovative Medicines Initiative 2 Joint Undertaking under grant agreement No. 875534. This Joint Undertaking support from the European Union's Horizon 2020 research and innovation programme and EFPIA and T1D Exchange, JDRF, and Obesity Action Coalition www.imisophia.eu. This communication reflects the author's view: neither IMI nor the European Union, EFPIA, or any Associated Partners are responsible for any use that may be made of the information contained therein. GH has received funding from the Innovative Medicines Initiative 2 Joint Undertaking under grant agreement No 875534. ARW is supported by the Academy of Medical Sciences / the Wellcome

Trust / the Government Department of Business, Energy and Industrial Strategy / the British Heart Foundation / Diabetes UK Springboard Award [SBF006\1134]. The research utilised data from the UK Biobank resource carried out under UK Biobank application number 9072. UK Biobank protocols were approved by the National Research Ethics Service Committee. The equipment utilised is funded by the Wellcome Trust Institutional Strategic Support Fund (WT097835MF), the Wellcome Trust Multi-User Equipment Award (WT101650MA), and the BBSRC LOLA award (BB/K003240/1). TMF is supported by MRC awards MR/WO14548/1 and MR/T002239/1. The authors would like to acknowledge the use of the University of Exeter High-Performance Computing (HPC) facility in carrying out this work, funded by an MRC Clinical Research Infrastructure award (MRC Grant: MR/M008924/1). The authors would like to gratefully acknowledge the studies and participants who provided biological samples and data for TOPMed. Full study-specific acknowledgements are detailed in the Supplementary Note.

## Author contributions

G.H., R.N.B., and M.N.W. conceived the study. G.H., R.N.B., Z.Li., R.M., and X.Li performed the analyses. G.H., R.N.B., Z.Li., R.M., X.Li, J.L., N.O. A.M., K.P., T.M.F., C.F.W., A.R.W., X.Lin, A.M., and M.N.W. wrote and edited the manuscript. C.M.A., D.K.A., A.E.A-K., A.A.A., K.C.B., E. B. J.A.B., A.P.C., N.C., Y.I.C., M.K.C., J.E.C., D.D., P.T.E., M.F., V.R.G., X.G., J.He., C.H., R.R.K., R.K., S.L.R.K., C.K., R.J.F.L., S.A.L., R.L. M., T.N., S.V., B.D.M., J.M.M., N.D.P., B.M.P., S.R., M.B.S., E.K.S., M.J.T., S.T.W., L.R.Y., H.Z., NHLBI Trans-Omics for Precision Medicine (TOPMed) Consortium, C.L., K.E. North and A.E. Justice contributed data, without which this study would not have been possible. All authors have read and approved the manuscript.

## Competing interests

Bruce M. Psaty serves on the Steering Committee of the Yale Open Data Access Project funded by Johnson & Johnson. Xihong Lin is a consultant of AbbVie Pharmaceuticals and Verily Life Sciences. The remaining authors declare no competing interests.

## Additional information

Gareth Hawkes [1,47] ✉, Robin N. Beaumont [1,47], Zilin Li [2,47], Ravi Mandla [3,47], Xihao Li [4,5,47], Christine M. Albert [6], Donna K. Arnett [7], Allison E. Ashley-Koch [8], Aneel A. Ashrani [9], Kathleen C. Barnes [10], Eric Boerwinkle [11], Jennifer A. Brody [12], April P. Carson [13], Nathalie Chami [14], Yii-Der Ida Chen [15], Mina K. Chung [16], Joanne E. Curran [17], Dawood Darbar [18], Patrick T. Ellinor [19], Myrian Fornage [11], Victor R. Gordeuk [20], Xiuqing Guo [15], Jiang He [21], Chii-Min Hwu [22], Rita R. Kalyani [23], Robert Kaplan [24], Sharon L. R. Kardia [25], Charles Kooperberg [26], Ruth J. F. Loos [14,27], Steven A. Lubitz [19], Ryan L. Minster [28], Take Naseri [29,30], Satupa'itea Viali [31,32,33], Braxton D. Mitchell [34], Joanne M. Murabito [35], Nicholette D. Palmer [36], Bruce M. Psaty [12,37], Susan Redline [38], M. Benjamin Shoemaker [39], Edwin K. Silverman [40], Marilyn J. Telen [41], Scott T. Weiss [40], Lisa R. Yanek [23], Hufeng Zhou [2], NHLBI Trans-Omics for Precision Medicine (TOPMed) Consortium, Ching-Ti Liu [42], Kari E. North [43], Anne E. Justice [44], Jonathan M. Locke [1], Nick Owens [1], Anna Murray [1], Kashyap Patel [1], Timothy M. Frayling [1], Caroline F. Wright [1], Andrew R. Wood [1], Xihong Lin [2,45,46], Alisa Manning [3] & Michael N. Weedon [1] ✉

[1]Clinical and Biomedical Sciences, University of Exeter, Exeter, UK. [2]Department of Biostatistics, Harvard T.H. Chan School of Public Health, Boston, MA, USA. [3]Department of Medicine, Harvard Medical School, Broad Institute, Boston, Massachusetts, USA. [4]Department of Biostatistics, University of North Carolina at Chapel Hill, Chapel Hill, NC, USA. [5]Department of Genetics, University of North Carolina at Chapel Hill, Chapel Hill, NC, USA. [6]Department of Cardiology, Smidt Heart Institute, Cedars-Sinai Medical Center, Los Angeles, CA, USA. [7]Provost Office, University of South Carolina, Columbia, SC, USA. [8]Department of Medicine, Duke Molecular Physiology Institute, Duke University Medical Center, Durham, NC, USA. [9]Division of Hematology, Department of Medicine, Mayo Clinic Rochester, Rochester, MN, USA. [10]Department of Medicine, School of Medicine, University of Colorado, Aurora, CO, USA. [11]Human Genetics Center, Department of Epidemiology, Human Genetics, and Environmental Sciences, School of Public Health, The University of Texas Health Science Center at Houston, Houston, TX, USA. [12]Cardiovascular Health Research Unit, Department of Medicine, University of Washington, Seattle, WA, USA. [13]Department of Medicine, University of Mississippi Medical Center, Jackson, MS, USA. [14]The Charles Bronfman Institute for Personalized Medicine, Icahn School of Medicine at Mount Sinai, New York, NY, USA. [15]The Institute for Translational Genomics and Population Sciences, Department of Pediatrics, The Lundquist Institute for Biomedical Innovation at Harbor-UCLA Medical Center, Torrance, CA, USA. [16]Department of Cardiovascular Medicine, Heart, Vascular & Thoracic Institute, Cleveland, OH, USA. [17]Department of Human Genetics and South Texas Diabetes and Obesity Institute, School of Medicine, The University of Texas Rio

Grande Valley, Brownsville, TX, USA. [18]Division of Cardiology, Department of Medicine, University of Illinois Chicago, Chicago, IL, USA. [19]Cardiovascular Research Center, Massachusetts General Hospital, Boston, MA, USA. [20]Department of Medicine, School of Medicine, University of Illinois at Chicago, Chicago, IL, USA. [21]Department of Epidemiology, Tulane University School of Public Health and Tropical Medicine, New Orleans, LA, USA. [22]Section of Endocrinology and Metabolism, Department of Medicine, Taipei Veterans General Hospital, Taipei City, Taiwan. [23]GeneSTAR Research Program, Department of Medicine, Johns Hopkins University School of Medicine, Baltimore, MD, USA. [24]Department of Epidemiology and Population Health, Albert Einstein College of Medicine, Bronx, NY, USA. [25]Department of Epidemiology, School of Public Health, University of Michigan, Ann Arbor, MI, USA. [26]Division of Public Health Sciences, Fred Hutchinson Cancer Center, Seattle, WA, USA. [27]Novo Nordisk Foundation Center for Basic Metabolic Research, Faculty of Health and Medical Sciences, University of Copenhagen, Copenhagen, Denmark. [28]Department of Human Genetics, University of Pittsburgh, Pittsburgh, PA, USA. [29]Naseri & Associates Public Health Consultancy Firm and Family Health Clinic, Apia, Samoa. [30]International Health Institute, Brown University, Providence, Rhode Island, US. [31]Oceania University of Medicine, Apia, Samoa. [32]School of Medicine, National University of Samoa, Apia, Samoa. [33]Dept of Chronic Disease Epidemiology, Yale University, New Haven, Connecticut, US. [34]Department of Medicine, University of Maryland School of Medicine, Baltimore, MD, USA. [35]Boston University's and National Heart, Lung, and Blood Institute's Framingham Heart Study, Framingham, MA, USA. [36]Department of Biochemistry, Wake Forest University School of Medicine, Winston-, Salem, NC, USA. [37]Departments of Medicine, Epidemiology, and Health Systems and Population Health, University of Washington, Seattle, WA, USA. [38]Division of Sleep and Circadian Disorders, Brigham and Women's Hospital, Boston, MA, USA. [39]Department of Medicine, Cardiovascular Medicine, Vanderbilt University Medical Center, Nashville, TN, USA. [40]Channing Division of Network Medicine, Department of Medicine, Brigham and Women's Hospital and Harvard Medical School, Boston, MA, USA. [41]Department of Medicine, Duke University School of Medicine, Durham, NC, USA. [42]Department of Biostatistics, School of Public Health, Boston University, Boston, MA, USA. [43]Department of Epidemiology, University of North Carolina at Chapel Hill, Chapel Hill, NC, USA. [44]Population Health Sciences, Geisinger, Danville, PA, USA. [45]Program in Medical and Population Genetics, Broad Institute of Harvard and MIT, Cambridge, MA, USA. [46]Department of Statistics, Harvard University, Cambridge, MA, USA. [47]These authors contributed equally: Gareth Hawkes, Robin N. Beaumont, Zilin Li, Ravi Mandla, Xihao Li. ✉e-mail: g.hawkes2@exeter.ac.uk; m.n.weedon@exeter.ac.uk

## NHLBI Trans-Omics for Precision Medicine (TOPMed) Consortium

Zilin Li[2,47], Ravi Mandla[3,47], Xihao Li[4,5,47], Christine M. Albert[6], Donna K. Arnett[7], Allison E. Ashley-Koch[8], Aneel A. Ashrani[9], Kathleen C. Barnes[10], Eric Boerwinkle[11], Jennifer A. Brody[12], April P. Carson[13], Nathalie Chami[14], Yii-Der Ida Chen[15], Mina K. Chung[16], Joanne E. Curran[17], Dawood Darbar[18], Patrick T. Ellinor[19], Myrian Fornage[11], Victor R. Gordeuk[20], Xiuqing Guo[15], Jiang He[21], Chii-Min Hwu[22], Rita R. Kalyani[23], Robert Kaplan[24], Sharon L. R. Kardia[25], Charles Kooperberg[26], Ruth J. F. Loos[14,27], Steven A. Lubitz[19], Ryan L. Minster[28], Take Naseri[29,30], Satupa'itea Viali[31,32,33], Braxton D. Mitchell[34], Joanne M. Murabito[35], Nicholette D. Palmer[36], Bruce M. Psaty[12,37], Susan Redline[38], M. Benjamin Shoemaker[39], Edwin K. Silverman[40], Marilyn J. Telen[41], Scott T. Weiss[40], Lisa R. Yanek[23], Hufeng Zhou[2], Ching-Ti Liu[42], Kari E. North[43], Anne E. Justice[44], Xihong Lin[2,45,46] & Alisa Manning[3]

A full list of members and their affiliations appears in the Supplementary Information.

