## [Peer Review File · Nature Communications]

Whole-genome sequencing in 333,100 individuals reveals rare non-coding single variant and aggregate associations with heightREVIEWER COMMENTS

Reviewer #1 (Remarks to the Author):

The manuscript by Hawkes et al. describes a rare noncoding variant discovery study for height using 200K participants from the UKB for discovery and 133K participants from TOPMed and All of Us for validation. The authors provide a nice template of how to conduct the analyses and identify 28 novel noncoding variants with high effect size compared to prior of which three were replicated in the validation cohorts.

Overall, this is a well-designed and written study with novel genomic associations of height beyond what has been discovered today from array-based GWAS. This study is limited by the sample size which the authors appropriately acknowledge and I anticipate and hope they would release a novel study once sample size doubles with the WGS data release on 500K UKB participants.

Specific comments:

- 1- For single variant analysis, unclear the rationale for the MAC greater than equal 20 cut-off and there is no mention of the rarity (eg. MAF <0.1%)
- 2- P-value cutoff for genomic aggregate association testing is not mentioned in the main text.
- 3- Figure 1 font is too small and illegible. This needs to be remade.
- 4- Figure 2 is a very nice illustration that the newly discovered 28 loci have much higher effect size compared to loci discovered from common variant GWAS. Consider labeling the red dots with the names.
- 5- The manuscript could be strengthened by developing a polygenic score of height that leverages the new associations discovered and testing its performance compared to prior scores based on common associations alone.

Reviewer #2 (Remarks to the Author):

Overall this was a well-done and thoughtful paper diving into WGS data from the UKB and other cohorts for the first time. Height is a reasonable first trait to try to dissect. The methods appear sound and the claims reasonable.

Major points:

I find the replication stats provided lackluster. "Ten of the 28 rare and low-frequency SNVs/indels we identified showed nominal ($p < 0.05$) evidence of replication in a meta-analysis of TOPMed and All of Us when we would expect 1-2 (1.4 expected at $P = 0.05$) – ST5." Please provide more detail on whether you were powered to replicate the remaining 18 signals, and how many carriers there were in the replication datasets and whether the betas were trending in the right directions. In other words, were these lack of replication or lack of power? Same for the lack of replication for the aggregate signal for PRR5-ARHGAP8. If these 28 signals were well powered for replication and still did not replicate then that is actually a serious issue for the claims of the paper and the results

would need to be backed off quite a bit or work would need to be done to figure out what QC problem was allowing so many false associations to pass the significance threshold.

For the analysis conditioned on known associations, were those known associations LD pruned first? If not, wouldn't it be a problem to include a large number of highly correlated variants as covariates?

For the 4 significant aggregate associations in the table, it seems that for each only 1 analysis method pushed the p-value to significance (1 for ACAT, 2 for burden, 1 for SKAT). How bad were the p-values for these regions using the other methods? Is there any concern that these signals may be an artifact of a low number of carriers and the specificities of the association testing method?

Figure 3 and 4: I'm not used to seeing a qq plot for the variants that went into an aggregate test. Usually with aggregate tests you'll include variants that were seen only 1 or 2 times, which don't produce great stats for a plot like this. Did your aggregate test use all variants or only ones that had at least 20 carriers?

I really liked this paragraph in the Discussion, and it directly addressed some of my concerns and makes me feel very encouraged that this was a thoughtful and well done study: "We chose to report genetic aggregate results after correcting for known variation only, despite some genes (e.g. HMGA1) containing genetic variants that were independently significant in our analysis. Although conditioning upon independent variants within the aggregates often decreased the strength of association, we do not interpret this as a suggestion that the association at the locus is driven entirely by a single variant. This is a topical point for rare variant analyses: at sufficiently high sample sizes, we predict that a large proportion of genetic variants within an identified genetic aggregate will be independently associated. We propose that this does not imply, however, that the association itself is not aggregate."

Discussion: "Our work further highlights the importance of adjusting for common variants in rare and low frequency variant discovery analyses to circumvent linkage-driven associations. Before adjustment for common variants, we observed rare and low frequency variants, which dropped to 80 (non-independent) after adjusting." What work was done to determine whether the original associations or the new ones discovered here were truly the more associated ones?

The methods describe a sliding window method but the results from the sliding window are not mentioned in the results section. I assume nothing was significant (which seems to be the case in figure 1), but this would be good to say or just leave out the sliding window mention entirely. It also might be that the proximal aggregate associations mentioned in the results are from the sliding window, but that is not clear the way it is currently written.

The wording for proximal and regulatory could also use more standardizing. It seems reasonably explained in the methods, but then the header "Multiple rare variants, and a common variant, form an allelic series in a regulatory region upstream of HMGA1, with substantial effects on height" I think is referring to a proximal region. Figure 1 also looks like there were no significant hits in the

regulatory regions but that's not stated clearly in the text.

Overall, it seems that the associations seen were in the proximal and coding regions analyzed and not in the parts that were farther from genes. Do the results argue against the need for WGS and instead support simply expanding out from the exons a little bit more than is currently done?

Minor points:

The methods describe that regenie was used but not whether the authors did both step 1 and step 2, or how many variants were used for step 1 if so.

Figure 1 is great.

The finding of the CNV downstream of SHOX is really interesting, it shows how useful it is to include CNV data and I loved reading this part.

Line 313 "C17or49" should be "C17orf49"

Discussion: "We observed no additional rare variant associations after adjusting for common variation, despite some recent claims to the contrary, although we did not explicitly test adjusting for a polygenic risk score, as the study suggested." This sentence should be removed. Adjusting for common variants should improve power for identifying rare variants, and it is somewhat stochastic whether that improvement will drive one of your signals over the significance threshold. Also in the present study the variants conditioned on were only the ones on the same chromosome, so you would not expect to gain as much power as you could by using variants across the genome.

The format of the pdf supplement is really not useful. It appears to be a very wide table of stats by ancestry split over many pages.

Response to reviewers: "Whole Genome association testing in 333,100 individuals across three biobanks identifies rare non-coding single variant and genomic aggregate associations with height"

We would like to thank the reviewers for a thoughtful and considerate review of our paper. We have responded below to each of the individual comment and have adjusted the manuscript accordingly.

REVIEWER COMMENTS

Reviewer #1 (Remarks to the Author):

The manuscript by Hawkes et al. describes a rare noncoding variant discovery study for height using 200K participants from the UKB for discovery and 133K participants from TOPMed and All of Us for validation. The authors provide a nice template of how to conduct the analyses and identify 28 novel noncoding variants with high effect size compared to prior of which three were replicated in the validation cohorts.

Overall, this is a well-designed and written study with novel genomic associations of height beyond what has been discovered today from array-based GWAS. This study is limited by the sample size which the authors appropriately acknowledge and I anticipate and hope they would release a novel study once sample size doubles with the WGS data release on 500K UKB participants.

Specific comments:

1- For single variant analysis, unclear the rationale for the MAC greater than equal 20 cut-off and there is no mention of the rarity (eg. MAF <0.1%)

A MAC cut-off of 20 was chosen following other publications in WGS analysis (e.g. PMID 36303018) and because data restrictions on All of Us stop researchers from publishing on single variants with MAC<20. We have added this reference into the text of the methods section to justify our choice of threshold, and to give the corresponding MAF in our data (5.46×10^{-5}).

2- P-value cutoff for genomic aggregate association testing is not mentioned in the main text.

We have added a sentence detailing the p-value cutoff for genomic aggregate testing

3- Figure 1 font is too small and illegible. This needs to be remade.

Apologies for this – we have re-generated Figure 1 and increased the font size

4- Figure 2 is a very nice illustration that the newly discovered 28 loci have much higher effect size compared to loci discovered from common variant GWAS. Consider labeling the red dots with the names.

We're glad that you appreciate Figure 2. We did consider adding closest-gene labels to each dot, but the nature of non-coding associations makes it difficult to state via which gene/pathway the association is acting. We thus felt it was better to leave the variants unlabelled.

5- The manuscript could be strengthened by developing a polygenic score of height that leverages the new associations discovered and testing its performance compared to prior scores based on common associations alone.

The aim of the manuscript was to identify new biology via association testing of rare non-coding variants, and fundamentally demonstrate the added value of WGS. Given our focus on rare variation, the additional heritability explained by the novel loci is low (within-sample variation explained by novel rare variants identified here is 0.0056). Additionally, the smaller sample size used in our analysis compared to the GIANT meta-analysis (200,000 vs 5.4 million) will mean that effect size estimates will be less precise in our analysis, reducing the variance explained by any score calculated based on our results.

Reviewer #2 (Remarks to the Author):

Overall this was a well-done and thoughtful paper diving into WGS data from the UKB and other cohorts for the first time. Height is a reasonable first trait to try to dissect. The methods appear sound and the claims reasonable.

Major points:

I find the replication stats provided lackluster. "Ten of the 28 rare and low-frequency SNVs/indels we identified showed nominal ($p < 0.05$) evidence of replication in a meta-analysis of TOPMed and All of Us when we would expect 1-2 (1.4 expected at $P = 0.05$) – ST5." Please provide more detail on whether you were powered to replicate the remaining 18 signals, and how many carriers there were in the replication datasets and whether the betas were trending in the right directions. In other words, were these lack of replication or lack of power? Same for the lack of replication for the aggregate signal for PRR5-ARHGAP8. If these 28 signals were well powered for replication and still did not replicate then that is actually a serious issue for the claims of the paper and the results would need to be backed off quite a bit or work would need to be done to figure out what QC problem was allowing so many false associations to pass the significance threshold.

We have added the detail that 22/27 single variants we were able to test for replication were sign consistent, equating to a binomial $P = 1.51e-3$. Following correction of the effect sizes estimated in our study to account for overestimation of effect sizes in discovery samples ('winner's curse') following PMID:37721937, we find that only nine of the single variants had power $> 80\%$ in the replication sample to see associations at a nominal p-value of 0.05. In our replication we saw 10 signals pass this threshold. Additionally, at a Bonferroni threshold of 1.85×10^{-3} , we identified 3 signals passing this threshold while there were 4 with power $> 80\%$ for replication. We believe these results demonstrate that our replication is in line with expectations and does not point to a significant number of false positives identified in our analysis. We have added a sentence to this paragraph in the manuscript to put these results in context given our power calculations.

The reason for not attempting replication for PRR5-ARHGAP8 in the non-UKB datasets was due to differences in annotations across cohorts: the AOU and TOPMed team did not annotate fusion transcripts – we have added this detail to the manuscript. In the UKB-AFR and UKB-SAS analyses, we did not observe evidence of replication ($\log_{10}p = 0.710$ and 0.345 , respectively) – we have now added this to the manuscript. This is not unexpected, however, given the substantially smaller

sample sizes for the UKB-AFR and UKB-SAS analyses.

For the analysis conditioned on known associations, were those known associations LD pruned first? If not, wouldn't it be a problem to include a large number of highly correlated variants as covariates?

Thank you for your comment – the variants used for conditioning were, in majority, variants from a genome-wide association study of height sourced from the most recent GIANT consortium paper, which applied the GCTA-CoJo algorithm to their summary statistics, resulting in a list of variants with conditionally independent associations for height (Methods: “we conditioned our analyses on 12,661 variants from the latest GIANT height consortium analysis of 5.4 million people based on imputed genotype array data, an exome-array analysis of height, and genome-wide significant ($P < 5 \times 10^{-8}$)”). While this may result in variants in partial LD, they will all have conditionally independent effects on height and therefore should not induce statistical issues.

For the 4 significant aggregate associations in the table, it seems that for each only 1 analysis method pushed the p-value to significance (1 for ACAT, 2 for burden, 1 for SKAT). How bad were the p-values for these regions using the other methods? Is there any concern that these signals may be an artifact of a low number of carriers and the specificities of the association testing method?

For the table, we chose the result with the lowest p value for that unique unit-annotation combination to simplify our results and these were the exact masks and tests that we attempted to replicate in All of Us and TOPMed. For example, the log₁₀p value of the GH1 proximal-aggregate unit was significant for the additive test (11.65), SKAT (9.75) and ADD-ACATV (9.46). Further, we did not observe significant association for the singleton masks (which are tested by default by regenie), suggesting that low-carrier count is not causing artefactual associations.

Figure 3 and 4: I'm not used to seeing a qq plot for the variants that went into an aggregate test. Usually with aggregate tests you'll include variants that were seen only 1 or 2 times, which don't produce great stats for a plot like this. Did your aggregate test use all variants or only ones that had at least 20 carriers?

The aggregate tests include all variants with MAF ≤ 0.1%, down to singletons. By default regenie's SKAT/ACAT aggregate testing, MAC < 10 variants are collapsed into a single burden association, which were then combined with the variants above this threshold under the SKAT/ACAT frameworks.

I really liked this paragraph in the Discussion, and it directly addressed some of my concerns and makes me feel very encouraged that this was a thoughtful and well done study: "We chose to report genetic aggregate results after correcting for known variation only, despite some genes (e.g. HMGA1) containing genetic variants that were independently significant in our analysis. Although conditioning upon independent variants within the aggregates often decreased the strength of association, we do not interpret this as a suggestion that the association at the locus is driven entirely by a single variant. This is a topical point for rare variant analyses: at sufficiently high sample sizes, we predict that a large proportion of genetic variants within an identified genetic aggregate will be independently associated. We propose that this does not imply, however, that the association itself is not aggregate."

Thank you for this comment.

Discussion: "Our work further highlights the importance of adjusting for common variants in rare and low frequency variant discovery analyses to circumvent linkage-driven associations. Before adjustment for common variants, we observed rare and low frequency variants, which dropped to 80 (non-independent) after adjusting." What work was done to determine whether the original associations or the new ones discovered here were truly the more associated ones?

The GIANT consortium meta-analysis included up to 5 million individuals, and the Regeneron results from Exome sequencing were on ~500,000 individuals, compared to our study in 200,000 individuals. Therefore, if we were to have identified a variant which was causal above and beyond that tagged by previous studies, the difference in statistical power between our study and the GIANT meta-analysis means that resolving these loci would not be possible in the majority of cases. We therefore chose to focus on identification of novel associations (primarily with rare variants not covered by previous imputation based GWASs) rather than fine-mapping.

The purpose of adjusting for previously identified loci was that, without such adjustment it is not trivial to define signals which are independent of known loci. We attempted to verify that our associations were not driven by the known loci by comparing associations both adjusted and unadjusted for nearby common signals, which suggested that our identified loci were independent from known loci. There was only one novel locus (chr12:27953832:SG:T:G) where the association was significantly attenuated when adjusted for known loci. As shown in the locuszoom plots for this locus below, it is close to a known locus, however the attenuation of the signal still suggests that it is independent of the common signal and a full fine-mapping analysis to resolve the association patterns is beyond the scope of the analysis presented here.

Association for chr12:27953832:SG:T:G in analysis not conditioned on known loci

Association for chr12:27953832:SG:T:G conditional on known loci

The methods describe a sliding window method but the results from the sliding window are not mentioned in the results section. I assume nothing was significant (which seems to be the case in figure 1), but this would be good to say or just leave out the sliding window mention entirely. It also might be that the proximal aggregate associations mentioned in the results are from the sliding window, but that is not clear the way it is currently written.

Apologies for the confusion – you are correct that the sliding window analysis did not identify any associations not found by proximal-regulatory aggregate testing. However, given that the aim of the paper is to present our methodology for non-coding aggregate testing, we would prefer to retain the method itself in the manuscript. We have, however, mentioned the sliding window results in our results section, as suggested.

The wording for proximal and regulatory could also use more standardizing. It seems reasonably explained in the methods, but then the header "Multiple rare variants, and a common variant, form an allelic series in a regulatory region upstream of HMGA1, with substantial effects on height" I think is referring to a proximal region. Figure 1 also looks like there were no significant hits in the regulatory regions but that's not stated clearly in the text.

Apologies for the confusion – we use the term proximal-regulatory to refer to 5kbp either side of the UTRs, and intergenic-regulatory to be outside these regions. You correctly point out that we did not identify any intergenic-regulatory associations: we have wording throughout to try to clarify where it previously may not have been clear enough or led to confusion.

Overall, it seems that the associations seen were in the proximal and coding regions analyzed and not in the parts that were farther from genes. Do the results argue against the need for WGS and instead support simply expanding out from the exons a little bit more than is currently done?

For our aggregate based analysis, most of the annotations we applied to the data were centred around coding regions, meaning that, while we attempted to examine intergenic regions with the sliding-window based analysis, we were biased toward seeing associations in regions close to known coding genes. We believe that as sample size increases, and more annotations for intergenic regions become available, we may start to see more associations in these regions. Additionally, in our single variant analysis, not all signals were in regions proximal to coding genes - the largest distance to a known gene is >250,000bp.

Overall, we feel that it would be premature to suggest that our results argue against the benefit of WGS and support simply expanding out from exons based purely on the results of this analysis. Additionally, limitations such as ~10% of coding genes being missed by WES data in UKB, along with variable coverage over other genes, means that the comparison between the benefits of WES and WGS require careful consideration of the benefits and drawbacks of both methods.

Minor points:

The methods describe that regenie was used but not whether the authors did both step 1 and step 2, or how many variants were used for step 1 if so.

Apologies for this oversight. We did perform Step1 with ~ 500,000 SNPs, according to the procedure set out in the regenie documentation. We have added text to the methods describing this.

Figure 1 is great.

The finding of the CNV downstream of SHOX is really interesting, it shows how useful it is to include CNV data and I loved reading this part.

Line 313 "C17or49" should be "C17orf49"

We have corrected this typographical error.

Discussion: "We observed no additional rare variant associations after adjusting for common variation, despite some recent claims to the contrary, although we did not explicitly test adjusting for a polygenic risk score, as the study suggested." This sentence should be removed. Adjusting for common variants should improve power for identifying rare variants, and it is somewhat stochastic whether that improvement will drive one of your signals over the significance threshold. Also in the present study the variants conditioned on were only the ones on the same chromosome, so you would not expect to gain as much power as you could by using variants across the genome.

This sentence has been removed from the manuscript.

The format of the pdf supplement is really not useful. It appears to be a very wide table of stats by ancestry split over many pages.

Apologies – this seems to be a function of the submission system. We had uploaded an Excel spreadsheet.

REVIEWERS' COMMENTS

Reviewer #1 (Remarks to the Author):

The authors addressed the comment appropriately and have improved the manuscript. Congratulations on this important work.

Reviewer #2 (Remarks to the Author):

the authors have responded to my comments adequately